# Non-Selective Beta-Blockers Decrease Infection, Acute Kidney Injury Episodes, and Ameliorate Sarcopenic Changes in Patients with Cirrhosis: A Propensity-Score Matching Tertiary-Center Cohort Study

**DOI:** 10.3390/jcm10112244

**Published:** 2021-05-21

**Authors:** Tzu-Hao Li, Chih-Wei Liu, Chia-Chang Huang, Yu-Lien Tsai, Shiang-Fen Huang, Ying-Ying Yang, Chang-Youh Tsai, Ming-Chih Hou, Han-Chieh Lin

**Affiliations:** 1Division of Allergy, Immunology, and Rheumatology, Department of Internal Medicine, Shin Kong Wu Ho-Su Memorial Hospital, No.95, Wen Chang Rd., Shihlin District, Taipei 111, Taiwan; pearharry@yahoo.com.tw; 2Institute of Clinical Medicine, National Yang Ming Chiao Tung University, No.155, Sec. 2, Linong St., Beitou District, Taipei City 112, Taiwan; lious123456@gmail.com (C.-W.L.); extraboy25@gmail.com (C.-C.H.); sfhuang.dr@gmail.com (S.-F.H.); cytsai@vghtpe.gov.tw (C.-Y.T.); 3School of Medicine, National Yang Ming Chiao Tung University, No.155, Sec. 2, Linong St., Beitou District, Taipei City 112, Taiwan; tp9310@yahoo.com.tw (Y.-L.T.); mchou@vghtpe.gov.tw (M.-C.H.); hclin@vghtpe.gov.tw (H.-C.L.); 4School of Medicine, Fu Jen Catholic University, No.510, Zhongzheng Rd., Xinzhuang Dist., New Taipei City 242, Taiwan; 5Division of Allergy, Immunology, and Rheumatology, Department of Medicine, Taipei Veterans General Hospital, No.201, Sec. 2, Shipai Rd., Beitou District, Taipei City 112, Taiwan; 6Division of Clinical Skills Training, Department of Medical Education, Taipei Veterans General Hospital, No.201, Sec. 2, Shipai Rd., Beitou District, Taipei City 112, Taiwan; 7Department of Medicine, Taipei Veterans General Hospital, No.201, Sec. 2, Shipai Rd., Beitou District, Taipei City 112, Taiwan; 8Division of Gastroenterology and Hepatology, Department of Medicine, Taipei Veterans General Hospital, No.201, Sec. 2, Shipai Rd., Beitou District, Taipei City 112, Taiwan

**Keywords:** non-selective beta-blocker, cirrhosis, infection, acute kidney injury, muscle wasting, sarcopenic

## Abstract

Background: Cirrhotic complications resulting from portal hypertension can be considerably reduced by non-selective beta-blockers (NSBBs); however, scarce studies have investigated therapeutic agents for other complications. We aimed to investigate the effects of NSBBs on common cirrhotic complications of infection, acute kidney injury (AKI), chronic renal function declination, and sarcopenic changes. Methods: Medical records of hospitalization for cirrhosis with at least a 4-year follow-up were analyzed and selected using propensity-score matching (PSM). Generalized estimating equation (GEE) was applied to assess the association of NSBBs with infection requiring hospitalization and AKI. Chronic renal function declination was evaluated by slope of regression lines derived from reciprocal of the serum creatinine level. The covariates of CT-measured skeletal muscle index (SMI) alterations were analyzed by generalized linear mixed model. Results: Among the 4946 reviewed individuals, 166 (83 NSBB group, 83 non-NSBB group) were eligible. Using GEE, Charlson comorbidity index, Child-Pugh score and non-NSBB were risk factors for infection; non-NSBB group revealed a robust trend toward AKI, showed no significant difference with chronic renal function declination of NSBB group, and was negatively associated with SMI alteration. Conclusion: Chronic NSBB use lowered the episodes of infection requiring hospitalization and AKIs, whereas non-NSBB was associated with sarcopenic changes.

## 1. Introduction

Cirrhosis is a deleterious multisystem condition, comprising the complications associated with portal hypertension (PH), such as ascites, variceal bleeding, spontaneous bacterial peritonitis (SBP), hepatic encephalopathy (HE) and hepatorenal syndrome (HRS) [1]. Moreover, in cirrhosis, intestinal bacterial translocation aggravates proinflammatory and profibrogenic reaction, and the resultant endotoxemia leads to immune dysfunction as well as antimicrobial resistance [2,3]; association between bacterial translocation and HE or renal dysfunction has been reported [3,4]. Owing to multiorgan disruption, cirrhosis usually heralds poor life quality in addition to less life expectancy, but the medical treatment remains an enormous unmet need [5].

Non-selective beta-blockers (NSBBs), including propranolol, carvedilol, nadolol, and timolol, improve hemodynamic parameters and impede the cirrhotic process [6]. Robust data support the clinical benefit of NSBBs in PH-related complications such as variceal formation or growth, bleeding or re-bleeding, and ascites; likewise, NSBBs prevent the development of liver decompensation [7]. NSBB-responders have fewer events of SBP and HE [8]. Furthermore, NSBBs reduce the incidence of hepatocellular carcinoma (HCC) in patients with cirrhosis because of various etiologies, and improve survival in cirrhotic patients with ascites or referral for liver transplantation [9,10,11,12].

Therapeutic options for treating other complications are relatively insufficient. For kidney injury, medical treatment has exclusively consisted of terlipressin in combination with albumin, specifically for patients with HRS, and volume replacement with albumin for patients with acute kidney injury (AKI) [13]. Less attention has been paid to the prevention or treatment of chronic kidney disease (CKD) in cirrhotic patients, even though CKD is accompanied by more acute-on-chronic liver failure (ACLF) events and higher mortality [14]. Recent literature indicates that CKD in patients with cirrhosis can be attributed to cardiorenal syndrome resulting from the activation of sympathetic nerve that can theoretically be blocked by NSBBs [15]. Similarly, few medication options other than antimicrobial agents are available with protective effects on infection in cirrhotic patients, which is currently the most common cause of mortality in cirrhosis, while NSBBs ameliorate systemic and splenic immune dysfunction in cirrhotic patients [16]. Therefore, whether NSBBs have clinical benefits in CKD or infection events is worth pursuing.

A growing corpus of research on sarcopenia has displayed an interrelation to manifold chronic diseases, and the higher prevalence of sarcopenia in patients with cirrhosis than that in the general population [17]. A recent meta-analysis revealed an association between sarcopenia and adverse clinical outcomes in cirrhotic patients, such as poorer survival rates and an increased risk of infection [18]. However, few studies have investigated the association between medication and sarcopenia in cirrhosis; besides, propranolol improves muscle synthesis at the hypermetabolic phase, which is compatible with cirrhotic status [19,20]. Hence, we aimed to investigate the effects of NSBBs on sarcopenic changes, renal dysfunction and infection events.

## 2. Methods

### 2.1. Study Subjects

In this retrospective longitudinal tertiary-center cohort study, the medical records of patients older than 18 years who were hospitalized for cirrhosis between 2006 and 2016 were reviewed. Patients who were regularly treated with NSBBs (including propranolol, carvedilol, nadolol, and timolol) and were followed up for at least 4 years were assigned to the NSBB-group, and those not using NSBBs were assigned to the control (non-NSBB) group. Patients treated with hepatobiliary surgery or liver transplantation, and those with a history of malignancy before enrolment, severe immunodeficiency/acquired immune deficiency syndrome, end-stage renal disease with renal replacement therapy, improper or undetermined diagnosis, inadequate NSBB administration (which was defined as occasional treatment without regular use, or discontinuation of NSBBs because of various reasons such as intolerance and non-responders) and the initiation of NSBB prescription before 2005 were excluded. The follow-up duration of each patient was four years retrieving from the available medical records. 

Basic demographic data on age, sex, etiologies of cirrhosis, date of cirrhosis diagnosis, type and initiation date of NSBB administration, and Child-Pugh scores were obtained. Comorbidities were recorded and the Charlson comorbidity index (CCI) was calculated. As to data abstraction, a preset stepwise protocol had been previously developed and reviewed by authors; three physicians and two nurses, who were familiar with the health records and trained in the data systems, retrieved the data according to the protocol step-by-step. All the processes abided by the methodological steps for retrospective chart review research as previously mentioned [21]. Furthermore, the recorded data were checked and confirmed by three physician authors. The protocol was approved by the Institutional Review Board of Taipei Veterans General Hospital.

### 2.2. Outcome Measurements

Serious infection episodes requiring hospitalization during the follow-up period were determined, and SBP was analyzed separately for its predisposition in patients with cirrhosis. A serious infection episode was objectively confirmed, defined as positive results of either microbiologic cultures or radiologic imaging, or both, in addition to corresponding antibiotics administration [22]. Plural infections at multiple sites at the same admission were counted as separate episodes if the previous one was treated completely. 

Renal dysfunction was assessed through the number of AKI episodes during the follow-up period. AKI was defined according to the Kidney Disease: Improving Global Outcomes (KDIGO) guideline as the presence of any of the following: (1) increase in serum creatinine (SCr) by ≥0.3 mg/dL within 48 h; (2) increase in SCr to ≥1.5 times baseline, which is known or presumed to have occurred within the prior 7 days; and (3) urine volume less than 0.5 mL/kg/h for 6 h. Furthermore, we calculated the decline of renal function by plotting the reciprocal of the serum creatinine level versus time, as previously reported [23]; a negative slope was considered as the progression of renal dysfunction. 

For skeletal muscle measurement, we evaluated the area of the total skeletal muscle (cm^2^) at the third lumbar (L3) level in computed tomography (CT) images through picture archiving and communication system, in which the skeletal muscles were identified by the Hounsfield unit threshold range of −29 to 150 at the corresponding site. The skeletal muscle index (SMI) was determined using the following formula: SMI = Area of total skeletal muscle (cm^2^) at the L3 level/ height squared (m^2^) [24]. 

### 2.3. Statistical Analyses

Concerning the potential baseline discrepancies between NSBBs and those in the non-NSBBs group, we conducted a 1:1 pair-matched case-control cohort by means of nearest-neighbor propensity-score matching (PSM), which was adjusted for age, sex, Child-Pugh score, CCI, and etiologies of cirrhosis after estimating the probability by logistic regression. Because liver diseases were included in the CCI, we deducted the points of relevant items while performing PSM, as previously described [25]. The pair-*t* and McNemar’s tests were used for continuous and categorical variables, respectively, when comparing the NSBB group versus the non-NSBB group. For estimating the episodes of infection and AKI, we performed the generalized estimating equation (GEE) to assess the relationship between NSBBs use and the occurrence of episodes after adjustment of the covariates such as age, sex, Child-Pugh score, and CCI. Differences in CKD between groups were examined from the slope of regression lines deriving from the reciprocal of the SCr level. A generalized linear mixed model (GLMM) with a mixed-effect model was used to investigate the covariates associated with SMI alteration after PSM. All *p*-values were subjected to two-sided tests; values <0.05 were considered significant. Data analyses were performed using Statistical Product and Service Solutions V.26 (SPSS, IBM, Armonk, NY, USA).

## 3. Results

As shown in Figure 1, the medical records of 4946 individuals had been elaborately reviewed, and 248 patients with well-documented 4-year data fulfilled our criteria; of these, 166 patients (83 NSBB users and 83 NSBB non-users) were eligible for analysis after PSM. Baseline demographics data are presented in Table 1, with mean age of 57.9 in the NSBB group and 56.5 in the non-NSBB group. In the NSBB group, 64, 16, and 3 patients were regularly treated with propranolol (average of 41.22 mg/day), carvedilol (average of 18.25 mg/day) and nadolol (average of 40 mg/day), respectively. No significant differences were observed between the groups in the baseline variables of age, sex, CCI, Child-Pugh score, MELD-Na score, serum creatinine and sodium level, presence of ascites and etiologies of cirrhosis (Table 1).

During the 4-year observation period, there were 34 independent infection episodes requiring hospitalization of 24 subjects in the NSBB group, and 50 episodes in 40 subjects in the non-NSBB group; three and four episodes of SBP occurred in the NSBB and non-NSBB groups, respectively (*p* = 0.719). According to the GEE model built on repeated measurement of infection episodes, CCI (odds ratio (OR): 1.467, *p* = 0.04), Child-Pugh score (OR: 1.175, *p* = 0.005) and those not using NSBBs (OR: 1.666, *p* = 0.035) posed significant risks to the infection requiring hospitalization; alternatively, age and sex did not influence infection (Table 2). 

Regarding AKI, 13 episodes of AKI were recorded in 10 NSBB users, and 25 episodes of AKI occurred among 20 subjects in the non-NSBB group within four years. The GEE analysis demonstrated that no use of NSBB led to a robust trend toward the occurrence of AKI (OR 2.070, *p* = 0.05, Table 3); otherwise, age, sex, CCI and Child-Pugh score were not associated with the AKI occurrence (Table 3). As for chronic renal function declination, the slope of the reciprocal of SCr concentration versus time in NSBB group was −0.0136 ± 0.0664 dL/mg/year, marginally lower than the counterpart, −0.0309 ± 0.0601 dL/mg/year in the non-NSBB group (*p* = 0.09, Figure 2).

Overall, 20 PSM pairs (40 patients) had abdominal CT profiles at the time of enrollment and the fourth year. The mean SMI alteration was 1.195 ± 5.633 cm^2^/m^2^ in the NSBB group, and −1.782 ± 4.624 cm^2^/m^2^ in the non-NSBB group. Using GLMM analysis with random effects of baseline SMI data, NSBB non-use was significantly associated with reduced SMI, compared to that in NSBB users (coefficient: −4.108, *p* = 0.049, Table 4); instead, age, sex, CCI, Child-Pugh score, episodes of infection/AKI, 4-year renal function declination, MELD-Na score, serum sodium and creatinine level, and presence of ascites were not correlated with SMI alteration (Table 4).

## 4. Discussion

In the present 4-year observational study, we investigated the relevant factors of infection requiring hospitalization, AKI and chronic renal function declination in patients with cirrhosis. Notably, the findings indicated that NSBB significantly decreased the infection, attained fewer AKI episodes, and marginally alleviated chronic renal function declination. Furthermore, sarcopenic changes in patients with cirrhosis were determined, and a negative correlation between SMI alteration and non-use of NSBBs was observed.

Owing to the widespread expression of the corresponding receptors on various immune cells, β-adrenergic signaling is considered as an essential immunomodulator [26]. In bacterial infection, β-adrenergic signaling suppressed inflammatory cytokines secretion, and immune response was centered by immune cells with additional apoptotic effects [27,28,29]; in experimental animal models, β-adrenergic blockade provided survival benefit and enhanced both cellular and humoral immunity against miscellaneous bacteria [30]. Similarly, propranolol corrected lymphopenia in cirrhotic mice [16]. Clinical studies have substantiated the beneficial results of the β-adrenergic blockade in patients with sepsis [31]; however, in other scenarios like stroke, the effects of β-blockers on protection from infection such as nosocomial pneumonia and urinary tract infection have been controversial [32,33]. 

In the context of cirrhosis, immune dysfunction including T-cell depletion and subset dysregulation is attributed to augmented sympathetic tone, mainly resulting from chronic endotoxemia following exaggerated intestinal permeability and bacterial translocation [34,35]. In previous studies, propranolol treatment decreased intestinal permeability and bacterial translocation, normalized the homeostasis and function of T cell subsets, ameliorated systemic immune dysfunction in cirrhosis, and increased phagocytic activity in the presence of bacteria [16,36,37]. Accordingly, NSBBs theoretically not only attenuate SBP, the major infectious complication stemming from bacterial translocation, but also mitigate the severity of systemic infection.

This study showed that, in line with the aforementioned deduction, in addition to age, Child-Pugh score and comorbidities, the absence of NSBBs (non-NSBB group) independently posed the risks of infection. Several studies have also reported lower rates of infection and alleviation of infection-related morbidity and mortality, as well as the reduction of likelihood of hospitalization for infection in cirrhotic patients, which was compatible with our findings [38,39]. In SBP, the infection directly related to intestinal bacterial translocation, albeit the data were debatable; previous studies and meta-analyses have indicated that NSBBs can prevent its occurrence and improve short-term survival in cirrhotic patients [12,40]. In the present work, an insignificant difference in SBP occurrence was observed between two groups, possibly due to relatively low incidence in the enrolled participants, although the proportions of SBP were approximate to those in previous study [39].

Hemodynamic aberrance in cirrhosis triggers activation of sympathetic tone and the renin-angiotensin-aldosterone system, leading to ischemia on kidney function [41]; β-adrenergic signaling blockade relieves the sympathetic tone and reduces serum renin level, consequently tempering resistance of renal blood flow and protecting against renal injury in cirrhotic patients [42,43]. However, insufficient blood volume resulting from the suppressed cardiac output and compromised arterial pressure may further deteriorate renal function, especially in decompensated cirrhosis or SBP [44]. Partially in line with the controversial roles of the β-adrenergic blockade in cirrhotic kidneys, foregoing studies have shown neutral results [45,46]. Our work yields a clear protective tendency of AKI occurrence by NSBB; one possible reason is comparatively less severity of the enrolled subjects, and NSBBs per se may prevent decompensation, thus manifesting the protective effects on AKI [7]. Evidence for the effect of long-term NSBB use in cirrhotic patients on chronic renal insufficiency is still limited; this 4-year study reveals marginal effects of deferring deterioration. 

Sympathetic overactivation in association with pro-inflammatory responses contributes to muscle wasting in cirrhosis; NSBB use can counteract sympathetic hyperactivity and ameliorate overwhelmed inflammation in patients with cirrhosis [16,47]. Additionally, sympathetic hyperactivity in hypermetabolic states, including burn injury and cirrhosis, can lead to muscle wasting [20,48]; NSBB use has been shown to attenuate catabolic muscle wasting in patients with burn injury, and furthermore promote anabolism in addition to enabling greater protein net balance [19,49]. In our study, the average SMI of patients with NSBB treatment increased, and non-NSBB was significantly negatively correlated with SMI, independent of other comorbidities and complications, all of which may be substantiated by the aforementioned literature.

Previous literature has also investigated the risk factors of skeletal muscle wasting in patients with cirrhosis, reporting that Child-Pugh scores were not significantly associated with muscle wasting; this is analogous to our findings [50]. Unlike the previous literature that analyzed Child-Pugh scores by grade, we evaluated the association between the levels of scoring and muscle wasting to derive more explicit results. Although previous studies have purported greater infection rates but insignificant higher infection-related mortality, little research has been conducted on the impact of infection on SMI; in our retrospective cohort, infection episodes were likely but not significantly associated with sarcopenic change [50,51]. 

Despite its designing as meticulously as possible, our retrospective observational study had some limitations. First, to explore the long-term effects and improve the comparability, we carefully reviewed the records and selected the cases using PSM, thus generating a relatively low case number; accordingly, we applied the statistical analyses proper for small sample sizes to reconcile the relevant bias. Second, the SMIs measured through CT imaging were recorded at a four-year interval rather than with a time-dependent evaluation, thus precluding the dynamic changes during NSBB use. Third, we defined the NSBB users as those whom were regularly treated with NSBB regardless of the dosage, which might have differed between individuals depending on their tolerance. Hence, well-designed prospective studies are warranted to elucidate more clear effects of NSBBs.

## 5. Conclusions

In conclusion, this study demonstrated the reduction of episodes of AKI and infection demanding hospitalization in cirrhosis by NSBBs; in addition, we also surfaced the mitigation of sarcopenic change in cirrhotic patients with NSBB treatment. We anticipate that this study may provide a basis of future comprehensive investigations on the effects of NSBBs in patients with cirrhosis. 

## Figures and Tables

**Figure 1 jcm-10-02244-f001:**
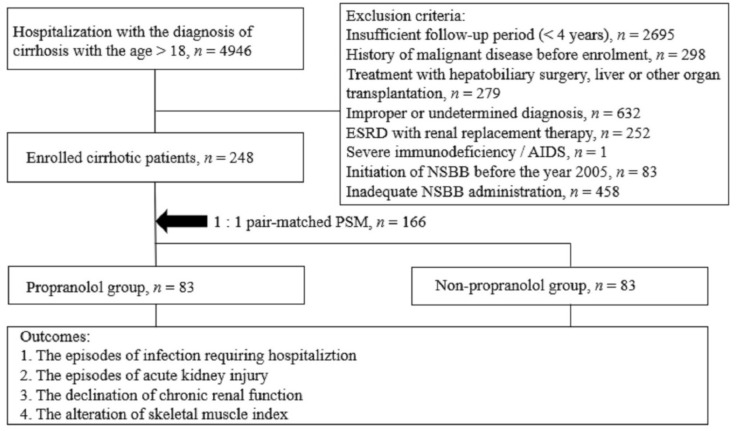
Algorithm of subject selection and enrollment. Abbreviation: AIDS, acquired immune deficiency syndrome; ESRD, end-stage renal disease.

**Figure 2 jcm-10-02244-f002:**
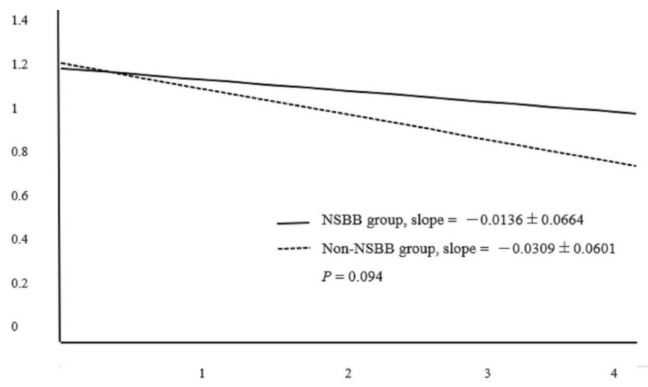
Comparison of the slope of the reciprocal of serum-creatinine concentration.

**Table 1 jcm-10-02244-t001:** Demographics of subjects in the NSBB and non-NSBB groups.

	NSBB (*n* = 83)	Non-NSBB (*n* = 83)	*p*-Value
Age (years)	57.9 ± 12.4	56.5 ± 12.1	0.468
Male (*n*)	49 (59.0%)	54 (65.0%)	0.511
CCI	2.71 ± 0.97	2.62 ± 0.79	0.683
CCI with deduction of points from liver disease	0.51 ± 0.632	0.47 ± 0.549	0.693
Child-Pugh score	6.90 ± 1.62	6.77 ± 1.56	0.598
Etiologies			
HBV	29 (34.9%)	30 (36.1%)	0.946
HCV	26 (31.3%)	25 (30.1%)	0.950
ALD	23 (27.7%)	28 (33.7%)	0.424
Autoimmune	5 (6.0%)	3 (3.6%)	0.625
Others	6 (7.2%)	6 (7.2%)	0.966
Presence of ascites	40 (48.2%)	46 (55.4%)	0.351
MELD-Na score	12.96 ± 3.86	13.22 ± 3.49	0.644
Serum sodium level (mmol/L)	119.39 ± 15.2	122.05 ± 13.25	0.162
Systolic blood pressure (mmHg)	138.01 ± 3.84	137.12 ± 4.33	0.233
Serum creatinine level (mg/dL)	1.01 ± 0.99	0.99 ± 0.56	0.883

Note: The data are expressed as the mean ± standard deviation or number (%). Abbreviations: ALD, alcoholic liver disease; CCI, Charlson comorbidity index; HBV, hepatitis B virus; HCV, hepatitis C virus.

**Table 2 jcm-10-02244-t002:** Factors associated with infection episode in patients with cirrhosis by generalized estimating equation.

	Odds Ratio	95% Confidence Interval	*p*-Value
Age	1.006	0.978–1.034	0.698
Female	0.984	0.556–1.742	0.955
CCI	1.467	1.019–2.114	0.04 *
Child-Pugh score	1.175	1.051–1.313	0.005 *
Non-NSBB	1.666	1.036–2.679	0.035 *
NSBB	Reference	-	-

* *p* value < 0.05 indicates statistical significance. Abbreviation: CCI, Charlson comorbidity index.

**Table 3 jcm-10-02244-t003:** Factors associated with acute kidney injury episode in patients with cirrhosis by generalized estimating equation.

	Odds Ratio	95% Confidence Interval	*p*-Value
Age	1.004	0.971–1.038	0.825
Female	1.590	0.868–2.913	0.133
CCI	0.865	0.493–1.520	0.814
Child-Pugh score	1.043	0.868–1.255	0.651
Non-NSBB	2.070	1.000–4.287	0.05
NSBB	Reference	-	-

Abbreviation: CCI, Charlson comorbidity index.

**Table 4 jcm-10-02244-t004:** Factors associated with skeletal muscle index alteration in patients with cirrhosis by generalized linear mixed model.

	Coefficient	95% Confidence Interval	*p*-Value
Age	−0.043	−0.233–1.417	0.507
Female	0.259	−4.374–4.891	0.909
CCI	−1.194	−4.663–2.275	0.486
Child-Pugh score	0.508	−4.663–2.275	0.446
Non-NSBB	−4.108	−8.204–−0.012	0.049 *
NSBB	Reference	-	-
Infection episode	1.839	−0.716–4.394	0.150
AKI episode	−0.407	−7.202–6.387	0.902
Slope of creatinine reciprocal	−8.107	−36.284–20.071	0.554
MELD-Na score	−0.235	−1.049–0.58	0.559
Serum creatinine level	−3.399	−12.03–5.231	0.427
Serum sodium level	−0.191	−0.871–0.488	0.568
Presence of ascites	−1.645	−22.62–19.332	0.873

* *p* value < 0.05 indicates statistical significance. Abbreviation: AKI, acute kidney injury; CCI, Charlson comorbidity index.

## Data Availability

The data presented in this study are available on request from the corresponding author.

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
