# Peer review of "Non-Selective Beta-Blockers Decrease Infection, Acute Kidney Injury Episodes, and Ameliorate Sarcopenic Changes in Patients with Cirrhosis: A Propensity-Score Matching Tertiary-Center Cohort Study"

_jcm, 2021, doi:10.3390/jcm10112244_

Round 1
Reviewer 1 Report
This is an interesting manuscript about the effects of non-selective beta-blockers on infection, acute kidney, ameliorate sarcopenic changes in patients with cirrhosis. This is a relevant topic with clinical implications. The manuscript is well written and limitations of the word (mainly its retrospective nature) are adequately addressed by the authors. However, there are several points that need to be clarified:
main issues:
1 - This is a retrospective study based on data recovery from clinical records. Did the authors have analyzed or know the quality of these clinical records?
2 - According to the nature of the manuscript, it is very relevant to assure the condition of patients of user or non-user of NSBB during 4 years. How did the authors control the adherence of patients to treatment during the period? is it possible to exclude that patients non-treated with NSBB had received occasional treatments during the period of time studied?.
3 - Dose of NSBB should be indicated.
4 - Figure 1: a total of 458 patients were excluded because inadequate NSBB administration. What is "inadequate NSBB administration"?
5 - on page 6, line 204, the expression "strongly attained fewer AKI episodes" may induce confusion because CI95% of AKI OR is very width and p value is 0.05. It would be better if the word "strongly" is removed from the sentence.
minor issues:
1 - on page 2, line 59 - the expression "including include" must be a mistake
2 - on page 3, line 136 - the expression "McNemur" must be a mistake
3 - on page 3, line 138 - the sentence "we performed has been paid he generalized estimating equation" should be reviewed
4 - on page 7, line 252 - is it right the expression "insufficiency if limited; this 4-year"?
Author Response
This is an interesting manuscript about the effects of non-selective beta-blockers on infection, acute kidney, ameliorate sarcopenic changes in patients with cirrhosis. This is a relevant topic with clinical implications. The manuscript is well written and limitations of the word (mainly its retrospective nature) are adequately addressed by the authors. However, there are several points that need to be clarified:
Main issues:
Point 1: This is a retrospective study based on data recovery from clinical records. Did the authors have analyzed or know the quality of these clinical records?
Response 1: Thanks for your constructive suggestion. In the present work, most of the outcome measurements were based on numerical data, which would be exempted from subjective judgement; besides, the episodes of infection requiring hospitalization adhered to the previously mentioned operational definition [Valdez-Ortiz R, et al. Int J Infect Dis. 2011;15:e188-96.]. As to data abstraction, a preset stepwise protocol had been previously developed and reviewed by authors; three physicians and two nurses, who were familiar with the health records and trained in the data systems, retrieved the data according to the protocol step-by-step. Furthermore, the recorded data were checked and confirmed by three independent authors (T.H. Li, Y.L. Tsai and Y.Y. Yang), who were hepatologist or physicians had trained in hepatology. All the processes abided by the methodological steps for retrospective chart review research as previously mentioned [Gearing RE, et al. J Can Acad Child Adolesc Psychiatry. 2006;15:126-34.]. According to your kind suggestion, we supplemented the relevant process in this revision (page 3, paragraph 3, line 107-111).
Point 2: According to the nature of the manuscript, it is very relevant to assure the condition of patients of user or non-user of NSBB during 4 years. How did the authors control the adherence of patients to treatment during the period? is it possible to exclude that patients non-treated with NSBB had received occasional treatments during the period of time studied?
Response 2: Thanks for your elaborate consideration. In the context of retrospective study, we reviewed the record carefully to rule out the cases interfering comparability; to ensure the adherence of treatment in our eligible cases, we excluded those with the outpatient follow-up interval longer than 6 months according to our available medical records. In the standardized medical chart of tertiary-center, all the recorders were obligated to record the concomitant medications, which could be obtained from history taking or the Taiwanese National Health Insurance PharmaCloud Systems, the platform displaying overall concurrent prescriptions of the patient [Liao CY, et al. Int J Med Inform. 2019;126:65-71]; therefore, the hepatologists mentioned whether NSBBs had been used in each cirrhotic cases via the platform, since all physicians and pharmacists were mandatory to check PharmaCould before prescription or administration of medication, and we accordingly confirmed the absence of NSBBs in the non-NSBB group.
Point 3: Dose of NSBB should be indicated.
Response 3: Thanks for your kind suggestion. In this revision, we supplemented the average daily dose of each NSBB (page 4, paragraph 1, line 158-159).
Point 4: Figure 1: a total of 458 patients were excluded because inadequate NSBB administration. What is "inadequate NSBB administration"?
Response 4: Thanks for your detailed review and we apologized for the vague expression. The “inadequate NSBB administration” referred to occasional treatment without regular use, or discontinuation of NSBBs because of various reasons, for example, intolerance, non-responders, etc. According to your kind suggestion, we added the text in this revision (page 2, paragraph 5, line 99-101).
Point 5: on page 6, line 204, the expression "strongly attained fewer AKI episodes" may induce confusion because CI95% of AKI OR is very width and p value is 0.05. It would be better if the word "strongly" is removed from the sentence.
Response 5: Thanks for your prudent consideration and we apologized for the arbitrary narration. According to your kind suggestion, we deleted the improper wording in this revision (page 6, paragraph 1, line 211).
Minor issues:
Point 1: on page 2, line 59 - the expression "including include" must be a mistake
Response 1: Thanks for your careful scrutiny. According to your kind suggestion, we deleted the redundant word in this revision (page 2, paragraph 2, line 59).
Point 2: on page 3, line 136 - the expression "McNemur" must be a mistake
Response 2: Thanks for your detailed review. According to your kind suggestion, we corrected the misspelling with “McNemar’s” in this revision (page 3, paragraph 6, line 141-142).
Point 3: on page 3, line 138 - the sentence "we performed has been paid he generalized estimating equation" should be reviewed
Response 3: Thanks for your thorough review and we apologized for the redundancy of wording. According to your kind suggestion, we deleted the superfluous wording in this revision (page 3, paragraph 6, line 144).
Point 4: on page 7, line 252 - is it right the expression "insufficiency if limited; this 4-year"?
Response 4: Thanks for your exhaustive review and we apologized for the confusion we had made. According to your kind suggestion, we rectified the misspelling, which we intended to express “is limited”, in this revision (page 7, paragraph 4, line 261).
We much appreciated your meticulous and comprehensive review to improve our manuscript.

Reviewer 2 Report
Ambitious 4 year longitudinal propensity score matching study comparing outcomes with patients with cirrhosis on NSBB with those not on NSBB. While groups were matched with CTP score, etiology of liver disease, sex, age and comorbidities, there is no mention of ascites, systolic blood pressure and baseline Cr and Na of these patients. By not including in PSM, concern that study is limited by comparing an inherently sicker population not able to tolerate NSBB clinically with patients that are less vasodilated with less HRS physiology able to tolerate NSBBs and also at less risk for decompensation. The study can be strengthened by including MELD-NA as well as ascites, Na and Cr in analysis. Sarcopenia often related to degree of ascites so again, would advocate including.
Author Response
Point 1: Ambitious 4 year longitudinal propensity score matching study comparing outcomes with patients with cirrhosis on NSBB with those not on NSBB. While groups were matched with CTP score, etiology of liver disease, sex, age and comorbidities, there is no mention of ascites, systolic blood pressure and baseline Cr and Na of these patients. By not including in PSM, concern that study is limited by comparing an inherently sicker population not able to tolerate NSBB clinically with patients that are less vasodilated with less HRS physiology able to tolerate NSBBs and also at less risk for decompensation. The study can be strengthened by including MELD-NA as well as ascites, Na and Cr in analysis. Sarcopenia often related to degree of ascites so again, would advocate including.
Response 1: Thanks for your prudent consideration and constructive recommendations.
In the context of retrospective study, we managed to rule out the cases interfering comparability, so we excluded those intolerable to NSBBs use and those discontinuation because of non-responding to NSBBs treatment, which partially indicating less vasodilated subjects; these cases were referred to “inadequate NSBB administration” in the present study, and we added the relevant instruction in this revision (page 2, paragraph 5, line 99-101). In addition, we analyzed MELD-Na, presence of ascites, serum sodium and creatinine levels, between NSBB and non-NSBB groups and also as the covariates of skeletal muscle index alteration; in this revision, we supplemented and amended the data according to your kind suggestion (page 4, paragraph 2, line 162; page 4, Table 1; page 5, paragraph 3, line 196-199; page 6, Table 4). We much appreciated your meticulous and comprehensive review to improve our manuscript.
